# Calibrated Spiking Messages for Emergent Multi-Agent Communication

## Abstract

We study emergent communication in multi-agent reinforcement learning (MARL) via a calibrated, bandwidth-aware framework that exchanges spiking messages built on a pretrained perceptual code. Agents share a spiking encoder (CommsMod) trained with a prototype–contrastive–sparsity objective, and use independent attention-based decision heads (DecisionMod) trained using calibration-aware Q-Learning. In referential games on Fashion-MNIST, with agents alternating sender/receiver roles, we assess protocol quality using within vs between class similarity, temporal attention consistency, and calibration; spike count serves as a bandwidth proxy. Experiments demonstrate the spiking channel yields accurate and sample-efficient communication, improves protocol discriminability, and reduces synaptic operations versus a matched continuous ANN baseline. Ablations show that (i) the shared pretrained encoder, (ii) temporal attention, and (iii) calibration terms are each necessary. Overall, semantically anchored, calibrated spiking communication offers a favourable accuracy–robustness–bandwidth trade-off and a practical route to neuromorphic deployment.

## 1 Introduction

Effective coordination in partially observable multi–agent environments often hinges on the emergence of useful communication protocols. A rich body of work has shown that agents can learn *what* to transmit, *when* to speak, and *whom* to address by training end–to–end through differentiable channels (Foerster et al., 2016; Lazaridou et al., 2017; Lazaridou & Baroni, 2020; Sukhbaatar et al., 2016; Iqbal & Sha, 2019; Das et al., 2019; Jiang & Lu, 2018). Concurrently, recent work has pushed scalability, structure, and interpretability with graph–based and hierarchical routing (Hu et al., 2024; Ding et al., 2024; Zhu et al., 2024), context–aware or personalised protocols (Li & Zhang, 2024), and language–grounded communication (Li et al., 2024), while new topologies target many–agent settings (Li et al., 2025).

Despite this progress, three limitations persist. First, messages are typically modelled as dense, real–valued vectors over ideal, bandwidth–free links (Foerster et al., 2016; Sukhbaatar et al., 2016); thus neither the *energy* nor the *temporal* structure of communication is represented explicitly, and robustness to noise and bit budgets is under–examined (Hu et al., 2024; Li & Zhang, 2024; Zhu et al., 2024). Second, most systems optimise for task return alone (Lazaridou & Baroni, 2020; Dagan et al., 2021) and do not assess *calibration*: are receivers appropriately confident in what they infer from messages, especially under distribution shift or channel impairments? Third, when perception and communication are learned jointly from scratch, sender and receiver may co–adapt idiosyncratically, making protocols brittle and limiting ad-hoc teamwork with new partners; recent studies on population training and independence highlight these risks and mitigation strategies (Michel et al., 2023; Pina et al., 2024).

We address these gaps by anchoring communication to a pretrained spiking perceptual basis and by jointly optimising calibration and return. Our framework couples a shared spiking encoder (CommsMod), pretrained with a prototype–contrastive–sparsity objective, with an attention-based decision head (DecisionMod). In MARL we freeze CommsMod so both agents share a stable spike lexicon; DecisionMod reads the sender's temporal spike train $S$ (from LIF dynamics with Poisson or latency coding) and learns via *calibration-aware deep Q-learning* under an explicit spike-

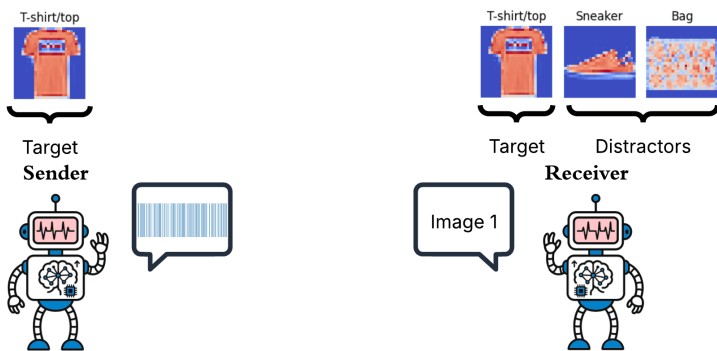

*Sender's CommsMod encodes the target into a spike train $S$.*

*Decodes with attention → scores$(Q, c)$; robust to spike budget$\|S\|_0$ & noise$\Phi_\eta$.*

Figure 1: **Referential Communication with Spiking Agents.** The sender (*Agent A*) encodes a target image with a pretrained (frozen) spiking encoder (COMMSMOD) into a temporal spike message $S \in \{0, 1\}^{T \times d}$ under a spike budget. The receiver (*Agent B*) encodes each candidate with its own COMMSMOD and uses an attention-based decision head (DECISIONMOD) to select the target and estimate confidence. Robustness is evaluated via a noisy channel operator $\Phi_\eta$ (drop/flip/latency) and bandwidth trade-offs.

count budget. Evaluation links message geometry to performance through protocol discriminability (within– vs between-class similarity gap), temporal-attention consistency, and standard calibration metrics (Expected Calibration Error, ECE; Maximum Calibration Error, MCE).

We ask whether (i) spiking messages under spike-count budgets rival continuous channels while being more robust to noise and bit limits; (ii) calibration-aware optimisation improves decision reliability without sacrificing return; and (iii) pretraining-and-freezing the shared encoder reduces co-adaptation and improves cross-play, and how architectural choices (temporal attention, coding scheme, temperature) affect discriminability and sample efficiency. Overall, the contributions of this paper can be summarised as follows:

1. *Calibrated spiking communication:* we propose a hybrid SNN–ANN framework in which agents communicate via LIF spike trains and DECISIONMOD is trained with a calibration-aware deep Q-objective (adaptive temperature + shaping).

2. *Semantically anchored protocols:* a pretrained COMMSMOD (prototype–contrastive–sparsity loss) shared and typically frozen across agents, yielding a stable spike-based codebook that curbs co-adaptation.

3. *Protocol-quality diagnostics:* we employ a compact suite—protocol discriminability, temporal-attention consistency, decision landscapes, and calibration metrics—that links message geometry to reward.

4. *Evidence under bandwidth and noise:* on Fashion–MNIST referential games as depicted in Fig. 1 (difficulty-aware curriculum), we observe high accuracy (e.g., ∼97% top-1 with $K{=}3$), strong discriminability, stable attention, and improved calibration versus ablations, with analyses across spike budgets and channel perturbations.

## 2   RELATED WORK

**End-to-end differentiable communication.** Early work established that agents can *learn* to communicate jointly with control. Foerster et al. (2016) introduced RIAL/DIAL, contrasting a reinforcement–only messaging scheme with a differentiable channel that permits gradient flow through messages. Sukhbaatar et al. (2016) proposed broadcasting continuous hidden states that are aggregated and fed back, improving coordination. Subsequent models refined *who/when* to talk via attention and routing (e.g., ATOC (Jiang & Lu, 2018), TARMAC (Das et al., 2019), AC-TOR–ATTENTION–CRITIC (Iqbal & Sha, 2019)) and scaled structure with graph/hierarchical de-

signs (Hu et al., 2024; Ding et al., 2024; Zhu et al., 2024), alongside language-grounded, human-interpretable channels (Li et al., 2024).

**Communication under noise and bandwidth constraints.** Most architectures idealise the channel; a complementary line embeds the *physical* link in the dynamics. Tung et al. (2021) jointly optimise control and communication over noisy multi-user channels, improving robustness over pipeline approaches. Context-aware gating and information-bottleneck views further motivate explicit budgets and stress-tests with dropouts, flips, and latency (Li & Zhang, 2024; Hu et al., 2024).

**Emergent language and referential games.** The language-emergence literature examines the *structure* and *interpretability* of learned codes in signalling/referential games: discrete protocols via straight-through/Gumbel relaxations (Havrylov & Titov, 2017), compositional structure and population effects (Lazaridou et al., 2017; Mordatch & Abbeel, 2018; Michel et al., 2023), and task settings ranging from pixel/symbolic referential games (Lazaridou & Baroni, 2020) to rules reasoning (Guo et al., 2023). Recent work sharpens evaluation of compositionality and one-to-many protocols (Lee et al., 2024; Carmeli et al., 2024), and probes generalisation in population-based training (Mu & Goodman, 2021; Verma, 2021). These diagnostics complement control-oriented MARL benchmarks and motivate our protocol-quality suite.

**Calibration and reliability.** Despite gains in return, communication–MARL rarely evaluates *calibration*. In single-agent deep learning, Guo et al. (2017) documented miscalibration and popularised temperature scaling and ECE/MCE. Bringing calibration to emergent communication is natural: miscalibrated receivers may over-trust messages, degrading robustness under shift or channel impairments.

**Spiking and neuromorphic perspectives.** Spiking neural networks offer sparse, temporally structured codes with attractive energy profiles on neuromorphic hardware (Davies et al., 2018). Surrogate gradients enable scalable training (Neftci et al., 2019), while event-based perception emphasises asynchronous, bandwidth-conscious computation (Gallego et al., 2022; Kugele et al., 2021; Negi et al., 2024). These insights motivate *spiking messages* as first-class communication primitives; however, explicit spike budgets and temporal coding remain under-explored in MARL.

We complement differentiable/attentional communication (Foerster et al., 2016; Sukhbaatar et al., 2016; Jiang & Lu, 2018; Iqbal & Sha, 2019; Das et al., 2019) with a neuromorphic and reliability perspective: a pretrained, shared spiking encoder to curb co-adaptation (cf. population effects Michel et al., 2023); explicit bandwidth via spike budgets and noisy-channel stress (Tung et al., 2021); and a calibration-aware objective with protocol-quality diagnostics aligned with emergent-language evaluation (Lee et al., 2024; Carmeli et al., 2024; Guo et al., 2017).

## 3 PROBLEM FORMULATION

[1] We cast emergent communication as a cooperative referential game in MARL. Each episode draws a candidate set $X = \{x^{(1)}, \ldots, x^{(K)}\} \subset \mathcal{D}$ (e.g., Fashion–MNIST); one element is sampled as the target, $x^\star = x^{(j^\star)}$ with $j^\star \in \{1, \ldots, K\}$.

**Agents and message space.** The sender $\mathcal{S}_\theta$ maps the target to a temporal spike message $S = \mathcal{S}_\theta(x^\star) \in \mathcal{M} \subseteq \{0,1\}^{T \times d}$, and the receiver $\mathcal{R}_\psi$ selects an index $a \in \{1, \ldots, K\}$ given the (possibly impaired) message and the candidates, $a = \mathcal{R}_\psi(\Phi_\eta(S), X)$. Here $\Phi_\eta : \{0,1\}^{T \times d} \to \{0,1\}^{T \times d}$ is a channel operator (e.g., drop/flip probabilities, latency jitter). Bandwidth/energy is exposed via a spike-count budget $\|S\|_0 = \sum_{t=1}^{T} \sum_{i=1}^{d} S_{t,i} \leq B$, enforced as a hard constraint or via a penalty.

**Perceptual factorisation.** Both agents share a pretrained spiking encoder $f_\phi$ (COMMSMOD): $\mathcal{S}_\theta(x) = g_\theta \circ f_\phi(x)$, $Z = \{f_\phi(x^{(k)})\}_{k=1}^{K}$. In most experiments $\phi$ is frozen, anchoring a stable spike vocabulary and reducing co-adaptation; $\theta, \psi$ are learned during MARL.

**Rewards and objective.** Episodes yield a sparse cooperative reward $r = \mathbb{I}\{a = j^\star\}$. During training we use a calibration- and bandwidth-aware signal $r_\lambda = r - \lambda_{\text{band}}\|S\|_0 - \lambda_{\text{cal}}\ell_{\text{cal}}$, with $(\lambda_{\text{band}}, \lambda_{\text{cal}} \geq 0)$. The receiver estimates state–action values $Q_\psi(\Phi_\eta(S), X, k)$ and calibrated

---

[1]Notation and tensor shapes are summarised in Appendix A.

confidence $c_\tau = \text{softmax}(Q_\psi/\tau)$ (temperature $\tau$). The optimisation is posed either as

$$\max_{\theta,\psi} \mathbb{E}[\,r\,] \quad \text{s.t. } \|S\|_0 \leq B, \tag{1}$$

or equivalently as the Lagrangian

$$\max_{\theta,\psi} \mathbb{E}[r - \lambda_{\text{band}}\|S\|_0 - \lambda_{\text{cal}}\ell_{\text{cal}}], \tag{2}$$

with calibration losses and training details in Section 4.

**Training protocol and evaluation.** We use a *similarity-graded distractor schedule* (a simple curriculum) that increases candidate hardness over training: distinct classes → mixed → visually similar variants. Roles alternate *per episode* (each agent acts as sender then receiver). Training is decentralised throughout: policies observe only local inputs and the exchanged message; we employ parameter sharing for the pretrained COMMSMOD, and target networks for stability (no centralised critic or global-state features). At evaluation time, execution is likewise decentralised; we optionally apply channel perturbations $\Phi_\eta$ (drop/flip/jitter) to $S$;

$$p = \min\Big(1, \tfrac{\text{epoch}}{E}\Big), \quad \pi_{\text{hardness}} = \big((1-p)^2,\ 2p(1-p),\ p^2\big), \tag{3}$$

where the three components weight sampling from *distinct*, *moderately similar*, and *very similar* class groups, respectively.

This setup lets us study: (i) effectiveness and robustness of *spiking* messages under $\Phi_\eta$; (ii) the effect of *pretraining/freezing* $f_\phi$ on partner generalisation (cross-play with unseen partners sharing $f_\phi$ but independent DECISIONMOD); and (iii) whether calibration-aware optimisation improves reliability without degrading return. We report sample efficiency as *episodes to a fixed communication success rate (CSR)* and area under the learning curve (AUC); protocol identifiability is assessed via variance of the within–between similarity gap across seeds.

# 4 METHODOLOGY

Our hybrid SNN–ANN framework comprises three tightly coupled parts: (i) a pretrained spiking encoder (COMMSMOD, Fig. 2) that maps images to temporally structured spike messages; (ii) an attention–based decision head with spiking integration and ANN readout (DECISIONMOD, Fig. 3); and (iii) a reinforcement–learning wrapper (SPIKEAGENT) that optimises task return, calibration, and a bandwidth proxy. The shared (typically frozen) COMMSMOD anchors communication to a stable spike vocabulary; DECISIONMOD learns to read these messages using temporal attention and hybrid SNN–ANN deep Q-learning.

## 4.1 COMMSMOD: SPIKING PERCEPTUAL ENCODER

In each episode the sender observes the target $x^\star$, and the receiver observes the $K$ candidates $X$. COMMSMOD transforms any image $x \in [0,1]^{H \times W}$ into a temporal spiking representation used as the message:

$$f_\phi: \ x \longmapsto \big(S(x) \in \{0,1\}^{T \times d},\ m(x) \in \mathbb{R}^d\big), \tag{4}$$

where $S$ is the spike train communicated by the sender (optionally perturbed by $\Phi_\eta$) and consumed by DECISIONMOD (Fig. 3) while $m$ is a normalised embedding used to pre-train the COMMSMOD.

**Spike encoders.** Over normalised pixels $x_i \in [0,1]$ and horizon $T$,

$$(\text{Rate/Poisson}) \ \ s_i^{(t)} \sim \text{Bernoulli}\big(x_i\,\tau(t)\,\Delta t\big), \ \ \tau(t) = 1 - 0.3\,t/T, \tag{5}$$

$$(\text{Latency}) \ \ t_i = \big\lfloor (1 - x_i)\,T \big\rfloor, \ \ s_i^{(t)} = \mathbb{I}[\,t \geq t_i \wedge x_i > \theta\,]. \tag{6}$$

Higher–intensity pixels fire earlier and/or more frequently, exposing a bandwidth proxy via spike count $\|S\|_0$ and temporal structure for attention. The same encoder is applied to target (sender) and to each candidate (receiver).

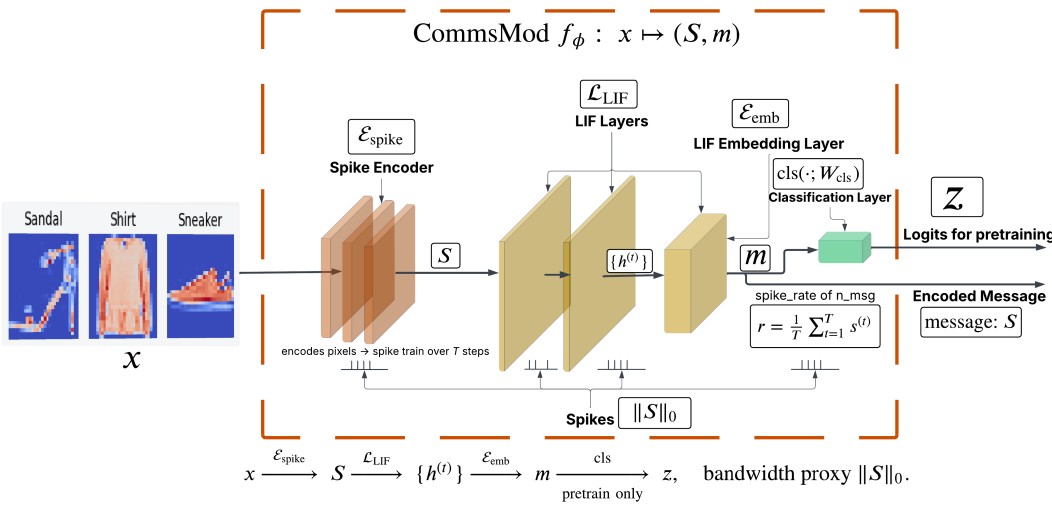

$$x \xrightarrow{\mathcal{E}_{\text{spike}}} S \xrightarrow{\mathcal{L}_{\text{LIF}}} \{h^{(t)}\} \xrightarrow{\mathcal{E}_{\text{emb}}} m \xrightarrow[\text{pretrain only}]{\text{cls}} z, \quad \text{bandwidth proxy } \|S\|_0.$$

Figure 2: **CommsMod.** Images are encoded to spike trains (rate *or* latency coding) and passed through LIF layers; an embedding layer aggregates activity and exposes a message vector $m$ (or the underlying spike train $S$) for downstream communication and pretraining classification.

**LIF dynamics and embedding.** Spikes $s_i^{(t)}$ traverse LIF layers with surrogate gradients (Fig. 2). With membrane potential $u^{(t)}$, decay $\beta \in (0,1)$, and threshold $v_{\text{th}}$,

$$u^{(t+1)} = \beta u^{(t)} + W s^{(t)} - v_{\text{th}} s^{(t)}, \qquad s^{(t)} = \Theta(u^{(t)} - v_{\text{th}}) \approx \tfrac{\alpha}{2} \operatorname{sech}^2(\alpha(u^{(t)} - v_{\text{th}})). \quad (7)$$

Embeddings combine spike rate and terminal membrane state,

$$e = 0.7 \frac{1}{T} \sum_{t=1}^{T} h_{\text{emb}}^{(t)} + 0.3 \, \sigma(u_{\text{emb}}^{(T)}), \qquad m = \frac{e}{\|e\|_2}. \quad (8)$$

**Pretraining objective.** COMMSMOD is pretrained with a prototype–contrastive–sparsity loss

$$\mathcal{L}_{\text{pre}} = \mathcal{L}_{\text{proto}} + \lambda_c \big( - \mathbb{E}_{(i,j) \in P} \big[ \cos(r_i, r_j) \big] + \mathbb{E}_{(i,j) \in N} \big[ \cos(r_i, r_j) \big] \big) + \lambda_s \, (\bar{s} - \rho)^2, \quad (9)$$

$$\mathcal{L}_{\text{proto}} = -\log \frac{\exp(e^\top p_y / \tau_p)}{\sum_c \exp(e^\top p_c / \tau_p)}. \quad (10)$$

Here $p_c$ are learnable class prototypes, $r_i = \frac{1}{T} \sum_t s_i^{(t)}$ is a spike–rate vector, $\bar{s} = \frac{1}{Td} \sum_{t,i} s_i^{(t)}$ the mean firing rate, and $\rho$ a target sparsity. The resulting checkpoint is shared by both agents and typically frozen during training to provide a stable spike vocabulary.

## 4.2 DECISIONMOD: TEMPORAL MATCHING AND READOUT

DECISIONMOD (Fig. 3) compares the sender's message to each candidate and outputs per–candidate values and calibrated confidence:

$$\big( S_{\text{s}}, \{S_{\text{c}}^{(k)}\}_{k=1}^{K} \big) \text{ or } \big( m_{\text{s}}, \{m^{(k)}\}_{k=1}^{K} \big) \longmapsto Q \in \mathbb{R}^K, \quad c = \operatorname{softmax}(Q/\tau) \in \Delta^K, \quad \alpha \in \Delta^T \quad (11)$$

Temporal attention produces $\alpha_t = \operatorname{softmax}(w^\top a_t + b)$ and an attention–pooled sender message $\tilde{s}_{\text{s}} = \sum_t \alpha_t s_{\text{s}}^{(t)}$ (and analogously $\tilde{s}_{\text{c}}^{(k)}$ for candidates). A message encoder yields compact features

$$f_{\text{s}} = \mathcal{M}_{\text{enc}}(\tilde{s}_{\text{s}}, m_{\text{s}}), \qquad f_{\text{c}}^{(k)} = \mathcal{M}_{\text{enc}}(\tilde{s}_{\text{c}}^{(k)}, m^{(k)}). \quad (12)$$

For each candidate $k$, cross–attention forms similarity features $z^{(k)} = \operatorname{CrossAttn}(f_{\text{c}}^{(k)}, f_{\text{s}})$, which are integrated by a spiking layer and read out:

$$Q_k = w_q^\top \left( \frac{1}{T} \sum_{t=1}^{T} \operatorname{LIF}_{\text{dec}}([f_{\text{s}}; f_{\text{c}}^{(k)}; z^{(k)}]) \right). \quad (13)$$

Here $\tau$ is a learnable temperature used for calibration (and, when sampling, exploration).

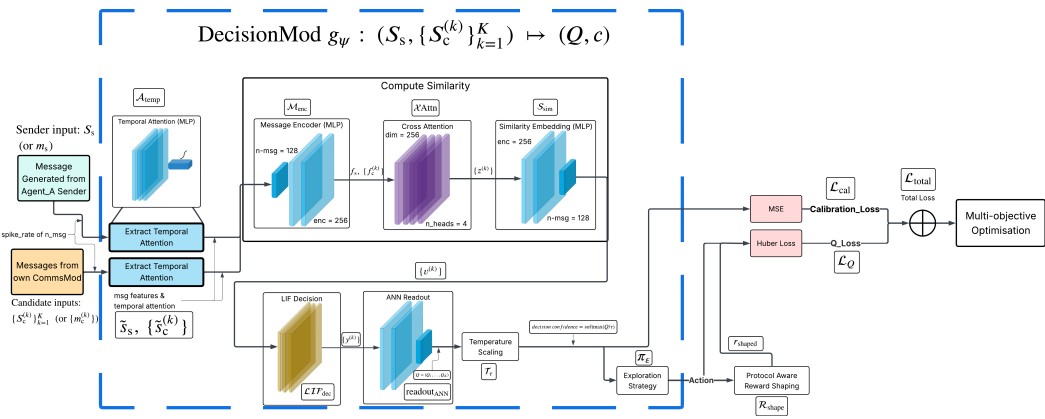

Figure 3: **DecisionMod.** Temporal attention summarises the sender's spike message and each candidate's encoding; a cross–attention block forms similarity features which are integrated by spiking layers and read out by an ANN to produce $Q$–values and calibrated confidence.

### 4.3 SPIKEAGENT: LEARNING WITH CALIBRATION AND BANDWIDTH

SPIKEAGENT wraps COMMSMOD+DECISIONMOD, and actions follow $(Q, c) \mapsto a$ (greedy, $\varepsilon$–greedy over $Q$, or sampling from $c$). We maximise expected return under a spike budget (or its Lagrangian):

$$\max_{\theta, \psi} \mathbb{E}[r] \quad \text{s.t.} \quad \|S\|_0 \leq B \quad \Longleftrightarrow \quad \max_{\theta, \psi} \mathbb{E}[r - \lambda_{\text{band}} \|S\|_0]. \tag{14}$$

The receiver minimises a calibration–aware TD objective:

$$\mathcal{L}_Q = \mathbb{E}\left[\text{Huber}\left(Q_a, \; r + \gamma \max_j Q'_j\right)\right], \qquad \mathcal{L}_{\text{cal}} = \|c_a - \tilde{c}_a\|_2^2, \tag{15}$$

where $\tilde{c}_a$ is a target confidence shaped by correctness (see Appendix B for the shaping rule and temperature–annealing schedule). The total loss adds entropy regularisation:

$$\mathcal{L}_{\text{total}} = \mathcal{L}_Q + \beta_{\text{cal}} \mathcal{L}_{\text{cal}} - \beta_H \, H\big(\text{softmax}(Q)\big). \tag{16}$$

Role–swapping (sender/receiver) and a *similarity-graded distractor schedule* over candidate similarity improve sample efficiency and stabilise protocol learning.

### 4.4 PROTOCOL QUALITY METRICS

We report task success and communication quality:

$$\text{CSR} = \mathbb{E}[\mathbb{I}[a = j^\star]], \qquad \delta = \underbrace{\text{sim}_{\text{within}}}_{\text{same class}} - \underbrace{\text{sim}_{\text{between}}}_{\text{different class}}, \tag{17}$$

with cosine similarity computed over normalised spike messages or embeddings. Temporal organisation is quantified by attention consistency $\text{Consist} = \frac{1}{1 + \text{std}_t(\alpha_t)}$, and decision reliability by standard calibration metrics (ECE/MCE). Bandwidth/energy is proxied by total spikes $\|S\|_0$.

Complete architectural and optimisation hyperparameters (including temperature annealing and ablation settings) are provided in Appendix B for exact reproducibility.

## 5 EXPERIMENTAL SETUP

**Task.** We evaluate a referential game with $K=3$ candidates per episode drawn from Fashion–MNIST (Xiao et al., 2017). The sender observes a target image and transmits a temporal spike message; the receiver selects the target using the shared (typically frozen) COMMSMOD and DECISIONMOD. Unless stated, we use a balanced subset of 1,000 images for protocol development and ablations; full–set runs are reported in the appendix. Spike trains use $T=25$ time steps ($\Delta t=1$ ms).

Table 1: Landscape of recent referential/emergent communication works (capabilities, not CSR).

| Work | Referential game | Channel | Bandwidth eval. | Calibration eval. | Co-adaptation mitigation |
|---|---|---|---|---|---|
| Lee et al. (2024) | ✓ | Discrete symbols | – | – | One-to-many structure |
| Carmeli et al. (2024) | ✓ | Discrete symbols | – | – | Compositionality diagnostics |
| Guo et al. (2023) | ✓ | Discrete symbols | – | – | Reasoning-focused setup |
| Michel et al. (2023) | ✓ | Discrete symbols | – | – | Population training, drift analysis |
| Li et al. (2024) | ✓[†] | Language tokens | – | – | Human-interpretable grounding |
| **This work** | ✓ | **Spiking (temporal)** | ✓ | ✓ | **Pretrained shared encoder** |

[†]Includes language-grounded multi-agent tasks with interpretable communication; settings vary. Capability flags reflect whether the cited work explicitly reports/optimises for the attribute.

**Model & training.** COMMSMOD employs 512–unit LIF layers with a 128–d embedding; DECISIONMOD uses 256–unit layers with 4–head cross–attention. LIF parameters: $\beta{=}0.9$, $v_{\text{th}}{=}0.8$. Optimiser: Adam ($1 \times 10^{-4}$), cosine decay (patience 10). Exploration decays $\epsilon$:$0.30 \rightarrow 0.02$ over 80% of training. Target networks soft–update with $\tau{=}0.005$. A shared replay buffer is used and roles swap per episode. Hardware: RTX 4090; throughput $\approx 2.7$ trials/s.

## 5.1 TRAINING PROTOCOL

**Phase 1 (pretraining).** COMMSMOD is trained with the prototype–contrastive–sparsity objective to 88% classification accuracy, yielding a sparse, class–separable spike code.

**Phase 2 (emergent communication).** Two SPIKEAGENTS load the shared COMMSMOD (frozen by default) and train DECISIONMOD with a calibration–aware deep Q–objective as demonstrated in Eq. (16). We employ: (i) *Similarity-graded distractor schedule* (distinct→mixed→visually similar) to harden candidates over training; (ii) *Adaptive temperature* driven by the accuracy–confidence gap with gentle annealing outside under/over-confidence regimes; (iii) *Protocol monitoring*: discriminability $\delta = \text{sim}_{\text{within}} - \text{sim}_{\text{between}}$, attention entropy/consistency, and calibration (ECE/MCE).

## 5.2 METRICS AND BASELINES

**Task performance.** Communication success rate (CSR; top–1 accuracy). **Protocol quality.** Discriminability $\delta$, within/between–class cosine similarities, attention consistency/entropy. **Calibration.** Reliability diagrams, ECE/MCE (5 bins, unless stated). **Bandwidth/efficiency.** Total spikes $\|S\|_0$ and relative synaptic operations (Appendix C for counting).

**Baselines.** (i) Continuous ANN channel with matched dimensionality and attention; (ii) discrete symbol channel (Gumbel–Softmax); (iii) ablations: no temporal attention, no calibration terms, no similarity-graded schedule, and frozen vs partially/fully finetuned COMMSMOD. Full grids and hyper-parameters are in Appendix C.

**Positioning vs recent referential/emergent works.** To avoid cross-dataset confounds, we compare capabilities rather than raw CSR with recent referential/emergent communication papers (Table 1). Numeric baselines (ours vs internal ablations *see Table 2*) are provided in Appendix D.5.

## 5.3 RESULTS

**Performance, efficiency, and the role of reward shaping.** As seen in Fig. 4, our hybrid spiking–ANN agent reaches $\approx 97.0\%$ CSR in the $K{=}3$ setting with rapid convergence (50% within 200 trials; 80% within 1,600; 90% within 2,400), and then stabilises. The right panel overlays the *base* task reward with the *shaped* reward used for training (base plus small bonuses for calibration and protocol quality; see. Section 4). The shaped signal (gold) consistently tracks, and slightly exceeds, the base reward (pink), yielding a denser learning signal, especially in early epochs, without overwhelming the task objective. This auxiliary guidance reduces variance, which is reflected in the left panel by the early crossing of the 90% line (epoch 24) and the narrowing confidence band as training proceeds. In short, reward shaping accelerates learning while preserving the qualitative ordering of policies implied by the base objective. These effects complement the efficiency benefits of the

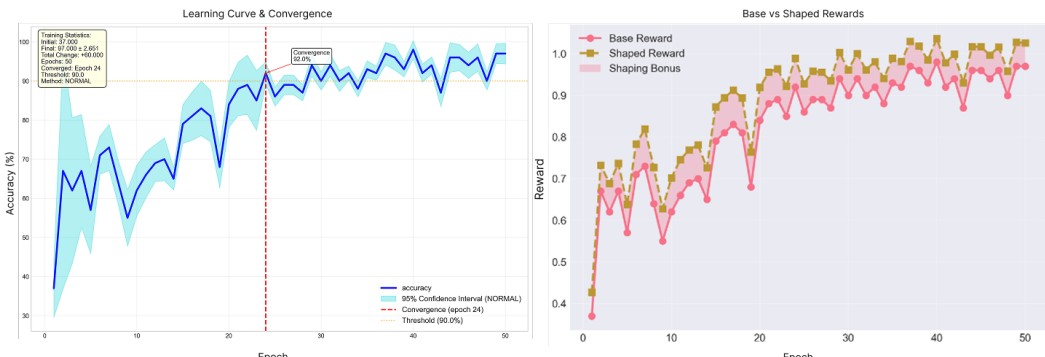

Figure 4: **Training dynamics.** (Left) CSR converges to ∼97% after 50 epochs reaching 92% by epoch 24. (Right) Protocol-aware shaping yields higher values by training completion.

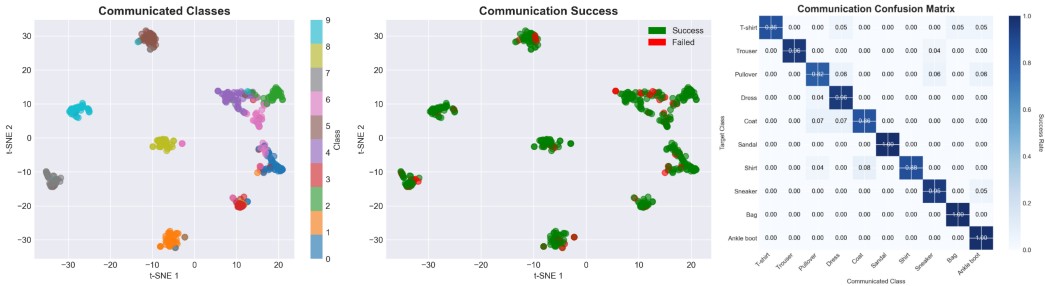

Figure 5: **Protocol structure.** (Left) t-SNE of spike-message embeddings shows class-organised clusters. (Centre) Trials are predominantly successful (green) with failures at cluster boundaries (red). (Right) Confusion matrix exhibits high diagonal accuracy.

Table 2: Compact comparison at matched bandwidth (Fashion–MNIST, $K=3$).

| Method | CSR (%) | Discriminability $\delta$ | ECE | Relative SynOps |
|---|---|---|---|---|
| Continuous ANN (vector channel) | 89.3 | 0.621 | 0.341 | 1.00 |
| Discrete symbols (Gumbel) | 86.5 | 0.588 | 0.322 | 0.92 |
| **Hybrid SNN–ANN (ours)** | **97.0** | **0.837** | **0.258** | **0.13** |

spiking channel (an $\sim 87\%$ reduction in relative synaptic operations versus a matched continuous ANN) and help explain the strong final performance reported in Table 2 and Appendix D.

**Protocol structure and temporal organisation.** Fig. 5 shows *semantically aligned* message clusters, failures concentrated at cluster boundaries, and a strongly diagonal confusion matrix (0.88–1.00), linking geometry to task success. The cosine gap reaches $\delta = 0.837$ (within $> 0.90$, between $< 0.10$), evidencing an *organised* protocol. Temporally, attention peaks near $t \approx 14$ with low spread and high consistency, indicating a shared "clock" that reduces effective temporal bandwidth and stabilises similarity. Table 2 corroborates this: removing temporal attention most degrades CSR and $\delta$ (and raises ECE), whereas freezing the shared COMMSMOD yields the most stable geometry at comparable accuracy.

**Calibration and decision quality.** Fig. 6 demonstrates that the calibration-aware head delivers *reliable* confidence. Mean confidence increases smoothly from $\approx 0.5$ to $\approx 0.65$ while accuracy reaches ∼95–97% (left), preserving a positive accuracy–confidence gap by construction of the adaptive temperature controller. The confidence–accuracy scatter (centre) follows a steep, near-linear trend approaching the identity but remaining below it, which is desirable: agents are not over-confident even when highly accurate. The reliability diagram (right) reports ECE = 0.258 and MCE = 0.322 (5 bins), indicating that most probability mass lies in bins where observed accuracy tracks predicted confidence. Practically, this matters for exploration and coordination: conservative confidence curbs

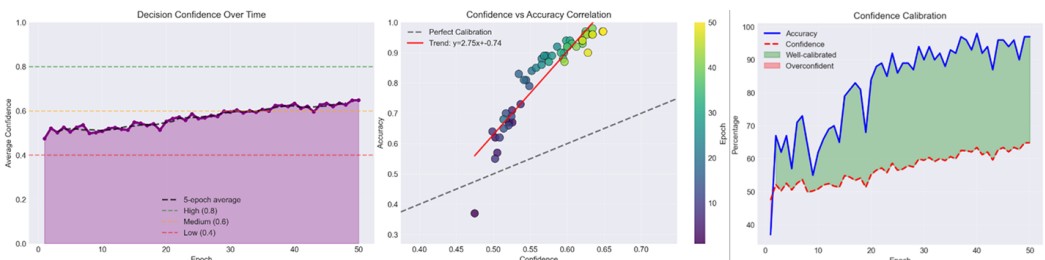

Figure 6: **Calibration.** (Left) Mean confidence rises from 0.5 to ∼0.65. (Centre) Confidence–accuracy correlation approaches the identity without overconfidence. (Right) Reliability diagram: ECE=0.258, MCE=0.322 (5 bins).

spurious exploitation and reduces error cascades when the sender/receiver face ambiguous candidates near cluster boundaries (as seen in Fig. 5).

**Comparisons and ablations (why each component matters).** The geometry and calibration results substantiate the architecture. Removing temporal attention weakens the mid-horizon focus, lowering discriminability $\delta$ and slowing convergence; consistent with attention concentrating readout on the informative spike window. Dropping calibration terms raises ECE without improving CSR, indicating the confidence head regulates certainty rather than trading reward for calibration. Disabling the similarity-graded schedule slows cluster consolidation and hurts early CSR, supporting a staged difficulty ramp. Freezing the shared COMMSMOD reduces sender–receiver co-adaptation, yielding lower variance in $\delta$ and steadier confusion diagonals than joint finetuning at similar accuracy. Table 2 corroborates these trends: each ablation consistently degrades CSR and $\delta$ while increasing ECE, with the largest hit from removing temporal attention, and the most stable geometry achieved when COMMSMOD is frozen.

# 6 CONCLUSIONS AND FUTURE WORK

We presented a calibrated, bandwidth–aware framework for emergent communication in multi–agent systems anchored to a pretrained spiking perceptual basis. Agents exchange temporal spike messages produced by a shared COMMSMOD and interpret them with an attention–based DECISIONMOD trained using a calibration–aware Q–objective. Beyond return, we evaluated protocol quality (discriminability and temporal attention) and decision reliability (ECE/MCE), linking message geometry and timing to downstream performance. On Fashion–MNIST referential games with $K=3$, the hybrid spiking–ANN agent achieved high CSR (∼97%), conservative calibration (ECE = 0.258), and a strongly organised message space ($\delta = 0.837$), while reducing relative synaptic operations by ∼87% compared to a *matched continuous ANN vector channel*. These results indicate progress on (i) explicit bandwidth/energy modelling via spike counts, (ii) reliability beyond accuracy via calibration–aware training, and (iii) reduced sender–receiver co–adaptation through a shared codebook.

Our study uses a single dataset (Fashion–MNIST), a fixed candidate set size ($K=3$), and a fixed random seed with controlled stochasticity; calibration is reported with 5-bin ECE/MCE. Channel perturbations are synthetic and message length fixed. These choices prioritise protocol analysis over breadth, and may limit external validity. Future directions for this work will involve (1) *Robustness and generalisation:* cross-play with heterogeneous partners, zero-shot transfer across $K$ and datasets, and harder distractor schedules. (2) *Communication structure:* multi-turn protocols, compression under explicit spike budgets, and learning adaptive budgets. (3) *Calibration and decision-making:* alternative calibration objectives, temperature schedulers, and decision-aware uncertainty aggregation across time. (4) *Hardware and efficiency:* mapping to neuromorphic substrates (e.g., SpiNNaker/Loihi), empirical energy/latency measurement, and selective finetuning/distillation of COMMSMOD for heterogeneous agents at the edge.

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

APPENDIX

## A  NOTATION AND SHAPES

**Shapes.** Unless stated, $f_{\mathrm{s}}$, $f_{\mathrm{c}}^{(k)}$, $z^{(k)}$ are projected to a common width $d_f$ before fusion; the concatenated similarity vector $\boldsymbol{f}_{\mathrm{sim}}^{(k)} = [f_{\mathrm{s}}; f_{\mathrm{c}}^{(k)}; z^{(k)}]$ has width $d_{\mathrm{sim}}$ (typically $3d_f$) and is the input to the LIF decision layer, which then aggregates over $T$.

Table 3: Symbols and tensor shapes. All vectors are column vectors unless stated.

| Symbol | Shape | Meaning |
|---|---|---|
| $H, W$ | – | Image height, width; $n = HW$ flattened pixels. |
| $K$ | – | Number of candidates per episode. |
| $T$ | – | Message horizon (time steps); $\Delta t$ step size. |
| $d$ | – | Message/embedding dimensionality. |
| $x \in [0,1]^{H \times W}$ | $H \times W$ | Input image (normalised); flattened to $\boldsymbol{x} \in [0,1]^n$. |
| $S = \{\boldsymbol{s}^{(t)}\}_{t=1}^T$ | $T \times d$ | Binary spike train; $\boldsymbol{s}^{(t)} \in \{0,1\}^d$ is time-$t$ spikes. |
| $\boldsymbol{m} \in \mathbb{R}^d$ | $d$ | $\ell_2$–normalised message embedding from COMMSMOD. |
| $\boldsymbol{e} \in \mathbb{R}^d$ | $d$ | Pre-normalised embedding (rate + membrane mix). |
| $u^{(t)} \in \mathbb{R}^{d_\ell}$ | $d_\ell$ | LIF membrane at layer $\ell$ and time $t$. |
| $f_\phi$ | – | COMMSMOD spiking encoder (shared; often frozen). |
| $f_{\mathrm{s}}$, $f_{\mathrm{c}}^{(k)}$ | $d_f$ | Sender/candidate features after message encoder in DECISIONMOD. |
| $z^{(k)}$ | $d_z$ | Cross-attention similarity features (candidate $k$ vs sender). |
| $\boldsymbol{f}_{\mathrm{sim}}^{(k)}$ | $d_{\mathrm{sim}}$ | Concatenated similarity features $[f_{\mathrm{s}}; f_{\mathrm{c}}^{(k)}; z^{(k)}]$. |
| $Q \in \mathbb{R}^K$ | $K$ | Per-candidate values; $\hat{Q}$ denotes normalised values. |
| $c = \mathrm{softmax}(Q/\tau)$ | $K$ | Calibrated confidence distribution. |
| $\alpha \in \Delta^T$ | $T$ | Temporal attention weights over the message horizon. |
| $\tau$ | – | Temperature (learnable; adaptively annealed). |
| $\beta, v_{\mathrm{th}}$ | – | LIF decay and threshold. |
| $\rho$ | – | Target sparsity (bandwidth proxy) in pretraining. |
| $\lambda_c, \lambda_s$ | – | Contrastive and sparsity weights in pretraining. |
| $\beta_{\mathrm{cal}}, \beta_H$ | – | Calibration and entropy weights in RL objective. |
| $\delta_H$ | – | Huber threshold; $\gamma$ discount; $\varepsilon$ exploration prob. |
| $\Phi_\eta$ | – | Channel operator (drop/flip/latency) with parameters $\eta = (p_d, p_f, J)$. |
| $B, \|S\|_0$ | – | Spike budget and total spike count (bandwidth proxy). |
| ECE, MCE | – | Expected/Maximum Calibration Error (with specified binning). |

## B  DESIGN RATIONALE AND PRACTICAL NOTES

**Why spiking messages?** Spikes expose *bandwidth and energy* through explicit counts ($\|S\|_0$) and carry *temporal structure* (latency/rate) that attention can exploit. This makes the channel physically meaningful under budgets, and naturally robust to certain perturbations (e.g., mild drop/jitter).

**Why a pretrained, shared COMMSMOD?** Freezing $f_\phi$ anchors sender/receiver to a *stable spike lexicon*, reducing co-adaptation and simplifying cross-play. The prototype–contrastive–sparsity objective yields class-separable, sparse codes that transfer cleanly to the communication game.

**Why hybrid DECISIONMOD?** Temporal attention highlights informative windows; cross-attention aligns sender and candidate features; a spiking integration layer retains temporal inductive bias, and an ANN readout stabilises gradients. Temperature-scaled softmax serves both *exploration* (when sampling) and *calibration* (at evaluation).

**Calibration objective.** Minimising $\|c_a - \tilde{c}_a\|_2^2$ encourages *appropriate* confidence (high when correct, low when wrong). Adaptive temperature keeps the accuracy–confidence gap positive (conservative) and avoids over-confident errors.

**Similarity-graded distractor schedule.** We harden the task by sampling candidates from distinct $\to$ mixed $\to$ very-similar class groups. This simple curriculum improves sample efficiency and yields more interpretable protocol geometry without changing the underlying MDP. Its effect is visible in Fig. 7 (stable high $\delta$) and Fig. 10 (consolidation of the attention peak).

**Decentralised training and execution.** Policies observe only local inputs and the exchanged message; we use parameter sharing for $f_\phi$, role swapping per episode, a shared replay buffer, and target networks. No centralised critic or global state is used.

**Bandwidth and efficiency proxies.** We report $\|S\|_0$ and relative *SynOps*. For SNNs, SynOps $\approx \sum_\ell (\#\text{spikes at layer } \ell) \times \text{fan\_in}_\ell$. For ANN references, MACs $\approx \sum_{\ell,t} (\text{in}_\ell \times \text{out}_\ell)$ over the same temporal extent, enabling like-for-like normalisation.

**Channel stressors.** $\Phi_\eta$ introduces drops ($p_d$), bit flips ($p_f$), and bounded latency jitter ($J$). We clamp indices to $[1, T]$ on jitter to preserve shape and avoid undefined timesteps.

## C IMPLEMENTATION DETAILS

- **Determinism.** All results use seed with determinism enforced for data order, initialisation, and RNGs used by Poisson sampling and exploration (report deviations if hardware libraries break determinism).

- **Normalisation.** Inputs are min–max normalised to $[0, 1]$ per split. Embeddings are $\ell_2$–normalised before similarity and cross-entropy with prototypes.

- **Stability constants.** Use small $\epsilon = 10^{-8}$ in normalisers (e.g., softmax, $\ell_2$, entropy) and clamp logits in reliability diagrams to avoid $\log 0$.

- **Jitter padding.** For $\Phi_\eta$ with jitter, index as $s_{\max(1,\min(T,t-J)),i}$ to keep message length fixed.

- **ECE/MCE.** Unless stated, 5 equal-width bins on $[0, 1]$; provide sensitivity to bin count in Appendix D.

- **Budgeting.** When enforcing a spike budget $B$, we either (i) penalise $\|S\|_0$ in the loss (Lagrangian) or (ii) hard-cap with rejection sampling during encoding—report which is used in each experiment.

- **Counting SynOps.** Count only triggered synapses (pre-spike $\to$ post multiply-accumulate). If you include reset/threshold ops, report separately.

- **Schedule.** The distractor schedule is parameterised by $p = \min(1, \text{epoch}/E)$ with sampling weights $((1-p)^2,\ 2p(1-p),\ p^2)$ for {distinct, mixed, very-similar}.

**Compute proxy.** *Relative SynOps* normalised by the continuous ANN reference; count attention/cross–attention for all methods as demonstrated the equations in Appendix D.5. Further hyperparameters (layer widths/depths, $\beta, v_{\text{th}}, \lambda_c, \lambda_s, \rho$, optimiser settings, temperature annealing constants, and ablation grids) are collated in Appendix B for exact replication.

# D  BASELINE PROTOCOL FOR TABLE 2

**Scope.** This appendix specifies, in a self–contained manner, how to reproduce the three rows of Table 2 at *matched bandwidth* on Fashion–MNIST ($K{=}3$, $T{=}25$). It covers task generation, bandwidth matching, architectures, training, metrics, evaluation, and reporting. Unless stated, we report **mean $\pm$ s.d. over 5 seeds** with deterministic flags enabled.[2]

## D.1  TASK AND DATA

**Dataset.** Fashion–MNIST (Xiao et al., 2017). Use a balanced 1,000–image subset for protocol development; full–set runs are reported in the appendix. Split 80/10/10 (train/val/test) with class–stratified indices saved to disk.

**Episode generation.** Each episode draws a target $x^\star$ and two distractors to form $K{=}3$ candidates. All methods use the same *similarity–graded distractor schedule* (distinct $\rightarrow$ mixed $\rightarrow$ very similar), parameterised by training progress as in Section 5. Roles swap *per episode* with probability 0.5.

## D.2  BANDWIDTH MATCHING (BITS PER EPISODE)

Let $S \in \{0,1\}^{T \times d}$ be the spiking message and $\|S\|_0$ its spike count. Define the episode bandwidth as

$$B \;=\; \mathbb{E}[\|S\|_0] \quad \text{(bits/episode)}. \tag{18}$$

We target a common $B_\star$ across methods with tolerance $\epsilon_B{=}1\%$ (relative).

**Continuous (vector) channel.** Uniform *per–tensor* affine quantisation to $b{=}8$ bits per scalar. Choose dimensionality $d_{\text{vec}}$ s.t.

$$T \cdot d_{\text{vec}} \cdot b \;\approx\; B_\star \quad \Rightarrow \quad d_{\text{vec}} = \left\lfloor \frac{B_\star}{8T} \right\rfloor. \tag{19}$$

Quantisation is applied during training and evaluation with straight–through rounding.

**Discrete (symbol) channel.** Choose alphabet size $L$ and tokens per episode $T_{\text{tok}}$ to satisfy

$$T_{\text{tok}} \log_2 L \;\approx\; B_\star, \tag{20}$$

with $T_{\text{tok}}$ chosen so that the effective message horizon aligns with $T$ for attention/cross–attention (pad or repeat tokens to length $T$ if needed).

**SNN budget tuning.** For spiking, we tune a Lagrange multiplier online to hit $B_\star$:

$$\lambda_{\text{band}} \leftarrow \lambda_{\text{band}} + \eta_B \left( \mathbb{E}[\|S\|_0] - B_\star \right), \tag{21}$$

with small step $\eta_B$ and clipping to $[0, \lambda_{\max}]$. We report realised $B$ on the test set and verify $|B - B_\star|/B_\star \le \epsilon_B$.

## D.3  ARCHITECTURES (IDENTICAL EXCEPT FOR THE CHANNEL)

**Ours (Hybrid SNN–ANN).** Pretrained COMMSMOD (LIF, 512 hidden, 128–d embedding), spike encoders (rate or latency), frozen by default; DECISIONMOD with 256–d blocks and 4–head cross–attention; calibration–aware Q–head as in Section 4. **Continuous ANN.** Replace spike message with $x_t \in \mathbb{R}^{d_{\text{vec}}}$ at each step; identical temporal attention, cross–attention, and decision head; 8–bit quantisation (straight–through) at train/test. **Discrete (Gumbel).** Sender emits $T_{\text{tok}}$ tokens; straight–through Gumbel–Softmax with temperature annealed 1.0→0.5; tokens embedded to 128–d and fed through the same attention and decision head. When needed, repeat/pad to length $T$ to keep attention costs *comparable*.

---

[2]If parts of the main text use a single seed, this is disclosed there; Table 2 uses 5 seeds.

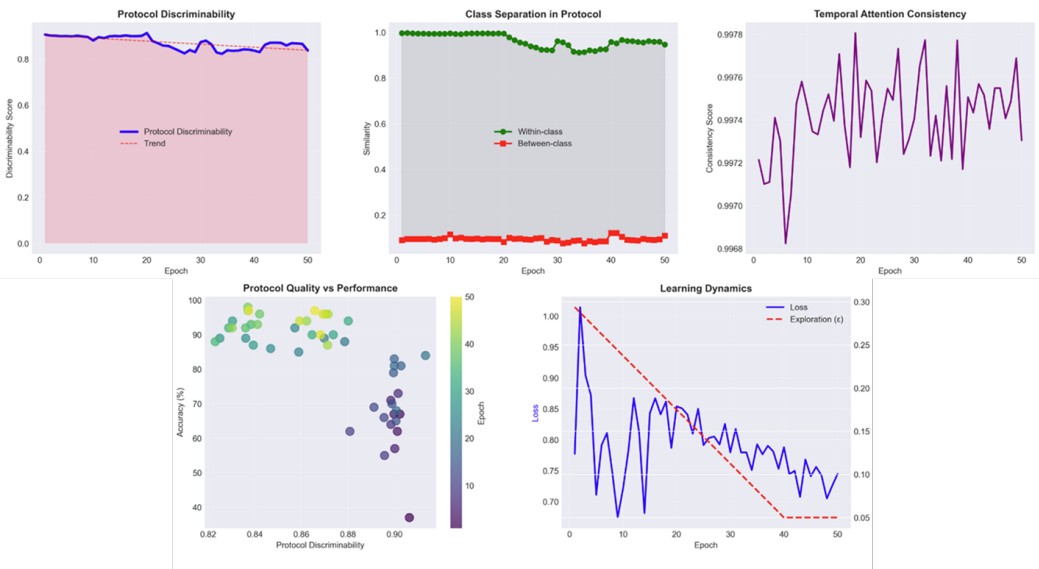

Figure 7: **Protocol dynamics.** (Top–left) Discriminability $\delta$ remains high and stable. (Top–middle) Clear separation: within–class similarity $\approx$ 0.95–1.0; between–class $\approx$ 0.06–0.10. (Top–right) Temporal attention consistency > 0.997 across epochs. (Bottom–left) Higher protocol quality correlates with task accuracy. (Bottom–middle) Loss decreases as exploration $\epsilon$ anneals.

### D.4 TRAINING DETAILS (ALL METHODS)

**Training paradigm.** *Parameter sharing with decentralised training:* shared replay buffer, target networks (soft update $\tau$=0.005), roles swap per episode. No centralised critic. **Optimiser.** Adam, lr $1 \times 10^{-4}$, cosine decay (patience 10). $\epsilon$-greedy from 0.30→0.02 over 80% of training. **Calibration.** Confidence target $\tilde{c}_a$ as in Section 4; adaptive temperature controlled by accuracy–confidence gap; identical scheduler across methods. **Budget.** Apply the $\lambda_{\mathrm{band}}\|S\|_0$ penalty (or the vector/discrete analogues via their bit counts) and tune $\lambda_{\mathrm{band}}$ to meet $B_\star$ within $\epsilon_B$. **Duration.** 50 epochs; retain the last–5 checkpoints. Unless noted, select the best validation checkpoint.

### D.5 METRICS DETAILS

**Task.** CSR = mean top–1 success over 5,000 test episodes. **Protocol discriminability.** $\delta = \mathrm{sim}_{\mathrm{within}} - \mathrm{sim}_{\mathrm{between}}$ with cosine similarity on *normalised* transmitted representations (SNN: spike–rate or embedding; vector: quantised vectors; discrete: token embeddings). **Calibration.** ECE/MCE using 5 equal–width bins on confidence; reliability diagram on the test set. **Bandwidth.** Report realised $B$ (bits/episode) and deviation $|B-B_\star|/B_\star$. **Compute proxy.** *Relative SynOps* normalised by the continuous ANN reference. Count attention/cross–attention and normalisation layers for *all* methods:

$$\mathrm{SynOps}_{\mathrm{SNN}} = \sum_{\ell} \big(\#\mathrm{spikes}_\ell \times \mathrm{fan\_in}_\ell\big), \tag{22}$$

$$\mathrm{MACs}_{\mathrm{ANN}} = \sum_{\ell,t} \big(\mathrm{in}_\ell \times \mathrm{out}_\ell\big), \tag{23}$$

$$\mathrm{Rel.\,SynOps} = \frac{\mathrm{Ops}_{\mathrm{method}}}{\mathrm{MACs}_{\mathrm{ANN\,ref}}}. \tag{24}$$

### D.6 EVALUATION AND REPORTING

Use 5 seeds; report mean $\pm$ s.d. for CSR, $\delta$, ECE, $B$, and Relative SynOps. State checkpoint selection (best–val vs last–5 average) in the caption. Release fixed splits, the seed list, the realised $B$, and code to verify the bandwidth tolerance $\epsilon_B$.

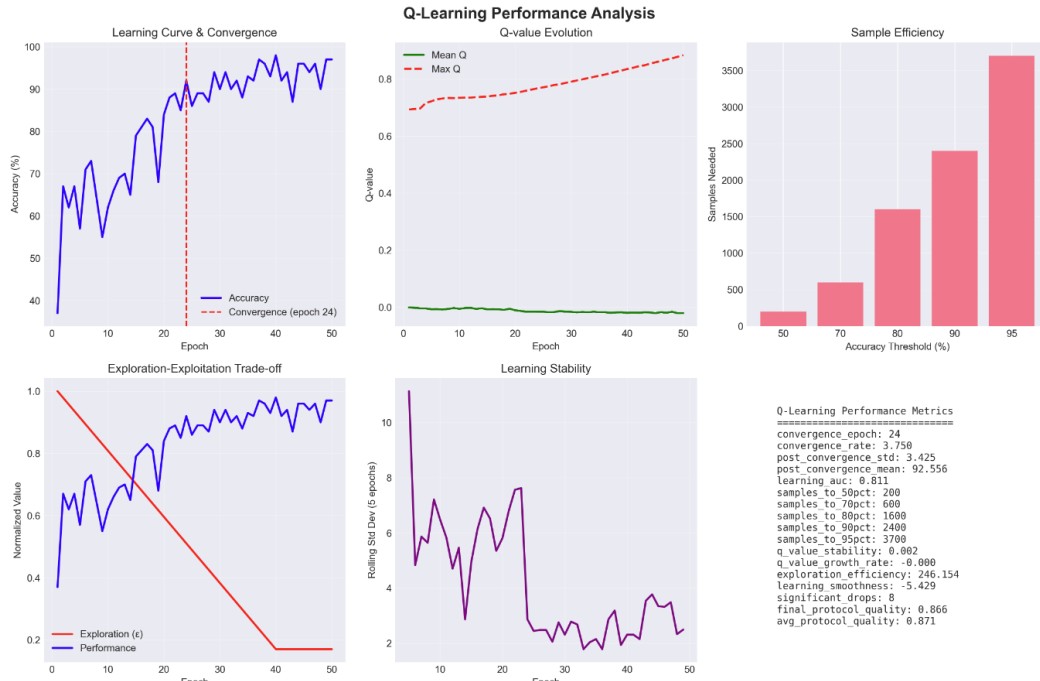

Figure 8: **Q-learning behaviour.** (Top–left) Accuracy curve. (Top–middle) Mean/Max $Q$ values remain bounded (normalisation in Section 4), avoiding blow-ups. (Top–right) *Sample efficiency*: approximate episodes needed to cross $50\%, 70\%, 80\%, 90\%, 95\%$ accuracy (numbers match the main text). (Bottom–left) Exploration–exploitation trade-off ($\epsilon$ vs performance). (Bottom–right) Rolling s.d. of accuracy indicates stabilising updates.

**System details (to be logged).** GPU model, driver/CUDA, PyTorch, cuDNN, OS; commit hash; all hyper–parameters; wall–clock time/epoch; deterministic/cuDNN flags.

# E    EXTENDED RESULTS AND VISUAL DIAGNOSTICS

This section complements the main findings with analyses that make our claims more transparent. Each diagnostic is motivated by Section 4, the metric definitions in Appendix D.5, and the objectives in Appendix B.

## E.1    PROTOCOL EVOLUTION AND TEMPORAL ORGANISATION

**Why these diagnostics?** Our protocol is carried by temporal spikes rather than dense vectors; we therefore need: (i) a *geometric* check that messages cluster by class (Discriminability $\delta$), and (ii) a *temporal* check that receivers read messages at consistent times (attention consistency). Fig. 7 shows that both hold: $\delta$ sits in the 0.83–0.90 band while attention consistency exceeds 0.997, indicating a shared readout "clock". The scatter (quality $\rightarrow$ accuracy) corroborates that improvements in message geometry transfer to task reward, justifying the use of $\delta$ as a proxy for protocol quality in Section 5.

## E.2    LEARNING BEHAVIOUR AND SAMPLE EFFICIENCY

**Why this matters.** The calibration-aware objective (Eq. ( 16)) adds terms beyond vanilla TD. Fig. 8 shows these additions do not destabilise learning: $Q$ trajectories stay well-behaved, and accuracy reaches the reported thresholds with few samples, supporting our claims of rapid convergence and sample efficiency.

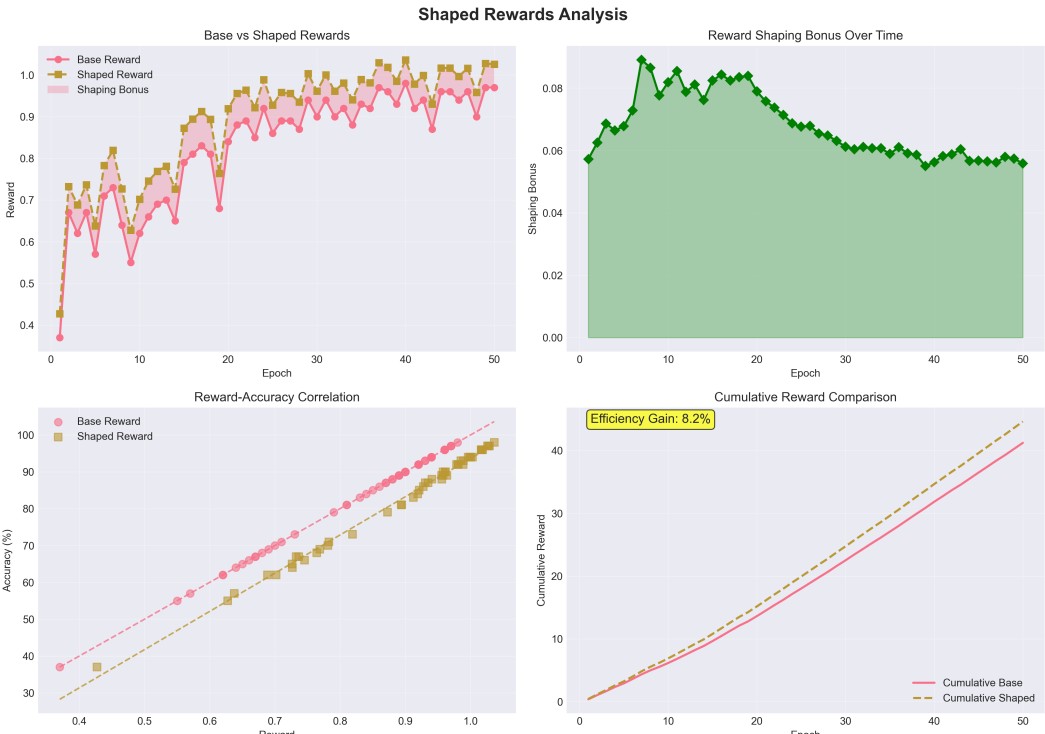

Figure 9: **Reward shaping.** (Top–left) Shaped reward (brown) tracks base reward (pink) with a small, consistent bonus. (Top–right) Bonus peaks early then decays, acting as a curriculum-like hint rather than a crutch. (Bottom–left) Reward–accuracy correlation remains monotone; shaping preserves training signal fidelity. (Bottom–right) *Cumulative* gain ($\approx 8.2\%$) evidences accelerated learning (improved sample efficiency) without overfitting to the bonus.

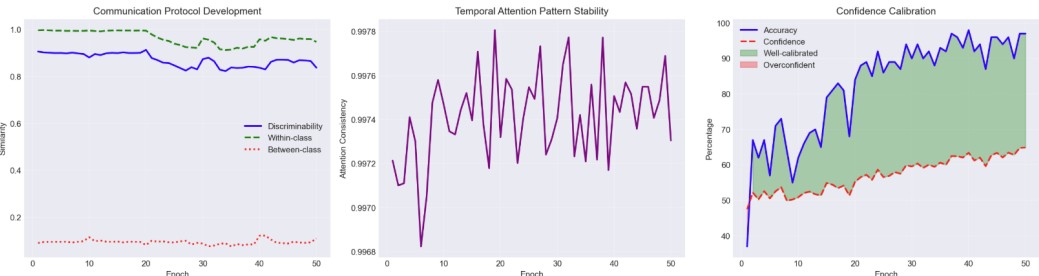

Figure 10: **Communication protocol, attention, and calibration.** (Left) Discriminability and within/between similarity curves. (Middle) Attention-pattern stability at the peak readout window. (Right) Accuracy stays above confidence throughout training (green band = well-calibrated regime).

### E.3 REWARD SHAPING ABLATION

Because the calibration and protocol terms in Eq. (16) augment the base task reward, we verify they improve *learning speed* rather than merely inflate the objective. Fig. 9 shows a transient bonus that accelerates early learning and then tapers off, while the reward–accuracy slope remains aligned—i.e., the agent still "earns" accuracy rather than relying on shaping. This justifies the inclusion of protocol-aware shaping in the main loss. Furthermore, cumulative shaped reward exceeds the base early and then grows in *parallel*, meaning the bonus mainly guides exploration at the start rather than inflating rewards throughout. In simple terms: shaping helps the agent learn the right behaviour *sooner* (higher area early on), but once competent it no longer gives an advantage; so performance gains reflect faster learning, not an artificial boost.

### E.4 Calibration and temporal readout

The temperature-scaled confidence (Section 4) is tuned by a homeostatic controller; Fig. 10 confirms it maintains a positive accuracy–confidence gap (no overconfidence) while the attention module locks onto a common time window, consistent with the latency/rate encoders. Together with the reliability scores in the main text (ECE/MCE), this supports our claim of *reliable* decision making.

### E.5 Compact protocol panels

**Reproducibility notes.** Unless stated, all plots were produced with the fixed seed described in Appendix D and the bandwidth–matched settings of Table 2. Scripts to regenerate the figures (train $\to$ eval $\to$ plots) are included in the released artefacts.

