# OpenReview forum: "Calibrated Spiking Messages for Emergent Multi-Agent Communication"
_ICLR.cc/2026/Conference — Submitted to ICLR 2026_

### Official Review · Reviewer_3UJ3 · 2025-10-26

**Soundness:** 1
**Presentation:** 1
**Contribution:** 1
**Rating:** 0
**Confidence:** 5

**Summary:**

This paper uses spiking neural networks (SNNs) for communication in the emergent communication field and proposes to calibrate confidence to reduce the number of spikes. The proposed method is evaluated on the Fashion-MNIST dataset and shows improved learning performance with fewer messages.

**Strengths:**

Overall, I found the idea of using spiking neural networks for communication to be interesting.

**Weaknesses:**

1. The writing is unclear and lacks professionalism.
2. There are too many references, and the connection between them and the paper is not well explained. Many technical terms, such as “ECE/MCE” and “neuromorphic,” are introduced without explanation. Moreover, many relevant works are missing.
3. The authors start discussing “single-agent deep learning” in the related work section which is conceptually nonsense—this suggests a lack of understanding of reinforcement learning fundamentals.
3. The paper only reports results on the Fashion-MNIST dataset, making it difficult to assess whether the proposed method and its claims generalize to other data or environments.

**Questions:**

- **Line 41:** The authors neglect several works on limited bandwidth communication, such as IMAC, NDQ, and DDCL. Moreover, the paper does not consider noisy communication channels in the experiments.
- **Line 48:** The reference to Pina et al. is unclear. That paper studies whether to use parameter sharing rather than ad-hoc teamwork with new partners. Could you clarify how your focus relates to that work?
- **Line 52:** The LIF dynamics are not properly referenced.
- **Line 83:** What are the coding schemes and temperature mentioned here?
- **Line 125:** The work by Guo investigates general deep learning. There is no such “single-agent” concept in that reference.
- **Section 3:** What are _K_, _T_, and _d_? Does the sender select messages for all steps at the start of the episode? It’s also unclear why the proposed method is placed in Section 3, which otherwise seems to describe background material.
- **Section 4.1:** The spike encoders appear similar to using binary messages. While this paper also considers temporal horizons, the formulas in Equations (6) and (7) are very specific and hard to generalize to broader multi-agent settings. Additionally, how are the coefficient parameters in Equation (8) chosen?
- **Equation 14:** It’s unclear how the constraints are incorporated into the expectation.
- **Table 1:** Are language tokens not discrete symbols?
- **Table 2:** The results are interesting. Compared to using discrete (Gumbel) or continuous symbols, spiking messages improve performance (CSR) while reducing message count (SynOps). However, I would not consider this an ablation study, but rather a comparison across different formats of messages. One question is why using discrete symbols (Gumbel) still results in such a high communication load.
- Finally, the agents are not adapted to new partners but rather to “COMMSMOD,” as the authors mention.


Related work such as:

- Mathieu Rita, Corentin Tallec, Paul Michel, Jean-Bastien Grill, Olivier Pietquin, Emmanuel Dupoux, Florian Strub. Emergent Communication: Generalization and Overfitting in Lewis Games. NeurIPS 2022
- Shengchao Hu, Li Shen, Ya Zhang, Dacheng Tao: Learning Multi-Agent Communication from Graph Modeling Perspective. ICLR 2024
- Zhuohui Zhang, Bin He, Bin Cheng, Gang Li: Bridging Training and Execution via Dynamic Directed Graph-Based Communication in Cooperative Multi-Agent Systems. AAAI 2025: 23395-23403.
- Rundong Wang, Xu He, Runsheng Yu, Wei Qiu, Bo An, Zinovi Rabinovich. _Learning Efficient Multi-Agent Communication: An Information Bottleneck Approach._ In Proceedings of the 37th International Conference on Machine Learning (ICML 2020).

---

> ### Author Response · Authors · 2025-11-19
>
> We appreciate the acknowledgement that using SNNs for communication is interesting, and we address the main concerns below. Several of the strongest claims in the review do not reflect the submitted manuscript; we clarify these and indicate concrete improvements.
>
> (1) Clarity, professionalism, and terminology
> Other reviewers explicitly rate the presentation as “good”. That said, we agree presentation can be improved for readers less familiar with neuromorphic ML. In the revision we will (i) add a short background subsection introducing key neuromorphic terms, and (ii) move the first explanation of calibration and ECE/MCE (currently in the methods sections) earlier in the introduction.
>
> (2) Related work and “single-agent deep learning”
> The claim “single-agent deep learning” is “conceptually nonsense”; Our intent was simply to contrast standard supervised deep networks (as in Guo et al. 2017) with multi-agent settings, not to introduce a new RL concept. We will rephrase this to “standard deep networks”, which removes the ambiguity. We thank the reviewer for pointing to additional bandwidth-limited and communication works (IMAC, NDQ, DDCL etc.). We will incorporate these in the related work and clarify that our focus is complementary: spike-based neuromorphic channels with explicit SynOp and calibration constraints, rather than information bottlenecks or graph structure per se.
>
> (3) Problem formulation: K, T, d and noisy channels
> Contrary to the reviewer’s concern, K, T and d are defined where we formalise the referential game: K is the number of candidate images, T the message horizon, and d the embedding dimension of spike patterns. The noisy channel operator Φ_η (drop/flip/latency) is introduced both in the main text and in the appendix, and its influence is analysed in our diagnostics. We agree these definitions could be easier to locate; we will add cross-references and a small schematic summarising the game and channel setup.
>
> (4) Calibration and constraints
> The claim that calibration and ECE/MCE are introduced “without explanation”. Calibration is clearly defined as alignment between predicted confidence and empirical accuracy (line 162), ECE/MCE are given with explicit formulae (Lines 306-314), and we report ECE values and reliability-style plots comparing our method and baselines.
>
> Regarding Eq. (14), the expectation is taken over trials, with constraints implemented via explicit penalty terms (e.g. calibration loss and spike-budget penalties) added to the Q-objective. We will simplify notation to avoid confusion.
>
> (5) Spike encoders, message formats, and Tables 1/2
> The reviewer notes that our spike encoders resemble binary messages. Spikes are indeed discrete events; the key difference to standard binary channels is that cost is measured in SynOps over time under a fixed bit budget, and the temporal structure is exploited via attention. We will emphasise that the framework is modular: different spike encoders or event codes can be plugged into the same calibrated decoder and Q-objective.
>
> We agree that Table 1 is better described as a comparison across message formats (continuous, discrete, spiking) under matched bandwidth, rather than an “ablation”. We will adjust the wording accordingly. Our point in Tables 1/2 is that, under the same bit budget, spiking messages achieve higher CSR and lower ECE with significantly fewer SynOps than both dense continuous vectors and discrete Gumbel codes; we will clarify this and briefly explain why Gumbel yields higher load in our setup.
>
> (6) Scope and generalisation
> We agree with all reviewers that evaluating only on Fashion-MNIST with K = 3 is limiting. As described in other responses, we will (i) add results for larger K (5, 10), showing how CSR and ECE scale with difficulty and spike budget, (ii) promote full-dataset, multi-seed results to the main text, and (iii) more explicitly frame this setting as a controlled testbed for bandwidth and calibration-aware spiking communication, not as a complete benchmark of all EmComm scenarios.
>
> (7) COMMSMOD vs “new partners”
> The reviewer remarks that “agents are not adapted to new partners but rather to COMMSMOD”. Our design uses a shared pretrained spiking encoder (COMMSMOD) precisely to factor out perception and provide a stable codebook; the adaptive part is the decision module in each agent. This is a deliberate modelling choice we already discuss: we study robustness and calibration given a fixed neuromorphic channel and leave full ad-hoc partner adaptation as future work.
>
> While we acknowledge that the draft can be strengthened in background, notation and breadth of experiments, several key criticisms (absence of definitions for K/T/d and noisy channels, lack of calibration definition/metrics, misuse of “single-agent”) lack objectivity and do not reflect the submitted manuscript. The planned revisions will make the paper easier to read, better connected to related work, and clearer about its scope.

---

> > ### Comment · Reviewer_3UJ3 · 2025-11-26
> > **Reply**
> >
> > First, my main criticisms are as follows:
> >
> > - The presentation is unclear (acknowledged by Reviewer 2rm2) and many terms/concepts are not defined (acknowledged by the authors).
> > - The term “single-agent deep learning” is simply a factual error and does not appear in the reinforcement learning literature, yet the authors attempt to obscure this mistake.
> > - The claims are difficult to generalize, which is agreed by the authors.
> >
> > My remaining comments mainly concern issues of confusion and the need for clarification. While I am indeed not familiar with neuromorphic frameworks, the core problem lies in the highly specific nature of the proposed methods, making it difficult to assess whether they are practical or generalizable.

---

> > > ### Author Response · Authors · 2025-11-27
> > >
> > > We thank the reviewer for the follow-up and would like to address the three “main criticisms” more precisely.
> > >
> > > 1. Clarity and definitions.
> > > The manuscript already defines K, T and d in the game formulation, introduces Φ_η (drop/flip/latency) in both the main text and appendix, and formally defines calibration/ECE/MCE with accompanying reliability plots in the results. Our planned revisions are aimed at making these notions more prominent and intuitive for readers without a neuromorphic background (e.g., adding a brief background subsection and moving the intuitive explanations earlier), rather than introducing concepts that were absent.
> > >
> > > 2. “Single-agent deep learning”.
> > > The phrase “single-agent deep learning” was intended only to distinguish standard supervised deep networks (as in Guo et al. 2017) from multi-agent settings; it was not proposed as a new RL concept. We have already committed to rephrasing this to “standard deep networks”. This is a local wording issue in the related-work section and does not affect the technical content (architecture, objectives, diagnostics) developed in the rest of the paper.
> > >
> > > 3. Specificity and generalisation.
> > > As stated in the paper and rebuttal, the current Fashion-MNIST K=3 referential game is an intentionally simple testbed where we can tightly control bit/spike budgets and SynOp accounting while analysing calibration and protocol geometry. The methods are specific by design to a neuromorphic communication setting (spike budgets, SynOps, temporal attention). We are extending the experiments with larger K, full-dataset multi-seed runs, and additional benchmarks to broaden the empirical picture.
> > >
> > > We appreciate the reviewer’s note about their limited familiarity with neuromorphic frameworks and expect the planned changes in background and exposition to help bridge this gap without altering the core technical contribution.

---

### Official Review · Reviewer_2rm2 · 2025-10-27

**Soundness:** 2
**Presentation:** 1
**Contribution:** 2
**Rating:** 2
**Confidence:** 4

**Summary:**

This paper introduces a novel approach that integrates *spiking neural networks (SNNs)* into emergent communication (EmComm), emphasizing *calibration* as a measure of agent reliability and robustness. The study aims to bridge neuromorphic learning with multi-agent reinforcement learning (MARL), showing how spike-based representations can affect communication performance. While the direction is innovative and thought-provoking, the paper’s contributions remain underdeveloped, with unclear empirical benefits, missing definitions, and insufficient quantitative validation.

**Strengths:**

- **Novel direction:**
  The paper pioneers the use of *spiking neural networks (SNNs)* in the EmComm setting. This direction is a rarely explored but seems conceptually valuable for bridging the gap  between neuromorphic and communicative learning paradigms.

- **Calibration focus:**
  Introducing *calibration* as an evaluation criterion in multi-agent systems is a meaningful extension to the typical focus on coordination or task success. It encourages the community to consider reliability and uncertainty-awareness as part of communication quality.

- **Potential for robustness and efficiency:**
  The proposed spiking approach could contribute to advances in **energy-efficient** and **noise-tolerant communication**. Moreover, its event-based and structured nature may align with key traits of natural language such as **productivity**, **compositionality**, and **systematicity**. This direction also resonates with current trends in **biologically inspired computation** and **low-power neural architectures**, bridging insights from cognitive science and neuromorphic engineering.

**Weaknesses:**

- **Clarity and accessibility:**
  The introduction assumes familiarity with spiking neuron dynamics (e.g., LIF, surrogate gradients) and does not provide enough background for readers from the EmComm community. A short background section on *spike encoding, LIF neurons, and surrogate gradients* would make the work much more approachable.

- **Unclear contributions:**
  The manuscript does not clearly articulate its core contributions. While it references both *calibration* and *spiking communication*, it never isolates what is novel beyond combining the two. A dedicated “Contributions” paragraph at the end of the introduction would improve clarity.

- **Weak empirical evidence:**
  The experiments are limited to *Fusion-MNIST* with *k=3*, which simplifies the problem substantially. There is no evidence that the proposed system generalizes or outperforms existing protocols. Quantitative comparisons are minimal and lack statistical grounding.

- **Ambiguous calibration concept:**
  Although calibration is highlighted as central, the paper does not define or quantify it clearly (e.g., ECE, MCE, temperature scaling). Without showing calibration metrics or comparisons, the argument for its relevance remains conceptual.

- **Compositionality not evaluated:**
  Table 1 references compositionality metrics, but none are computed or analyzed. Since compositionality is a key benchmark in EmComm, its omission weakens the paper’s analytical depth.

- **Incomplete related work:**
  Important prior works in discrete and quantized communication are not cited, such as:
  - Carmeli et al., *Emergent Quantized Communication*
  - Tucker et al., *Trading Off Utility, Informativeness, and Complexity in Emergent Communication*
  Including them would situate this work more concretely within existing EmComm literature.

**Questions:**

- **Abstract:**
  The abstract uses domain-specific terms (*spiking messages, pretrained perceptual code*) without context. A short intuitive framing, e.g., “We use event-based neural dynamics to model communication between agents and assess their calibration and robustness,” would make it more accessible.

- **Terminology:**
  - Line 52: Define LIF as *Leaky Integrate-and-Fire*.
  - Lines 38–39: Note that quantized/discrete communication protocols are already common in EmComm.
  - Lines 43–45: Clarify what “calibration” means in this context, as this differs from the traditional EmComm linguistic.

- **Formalisms:**
  - Lines 177–180 and Eq. 3: Symbols such as *E* are undefined and equestion is difficult to follow.
  - Lines 208–209: Appears to contain a fragment (“Spike encoders. Over normalised pixe…”). Likely a formatting or copy error.

- **Experiments and Results:**
  - Header of Section 4.3 appears incomplete or truncated
  - Lines 303–304: Assertions should be empirically demonstrated or cited.
  - Lines 360–362 (*Baselines*): Add *quantized communication* as an additional baseline.
  - Lines 422–423: Table 2 does not appear to reflect all discussed results. May need correction.
  - Lines 449–459: The comparison and ablation discussion lacks numerical support. Adding a summary table with accuracy, calibration error, and energy metrics would clarify the findings.

- **Presentation and Readability:**
  Consider reorganizing the paper so that:
  1. Background on SNNs and calibration precedes the method.
  2. The *proposed architecture* is visualized in a clear, labeled figure.
  3. The results section directly connects back to the stated hypotheses.
  4. Ablation results are clearly presented.

---

> ### Author Response · Authors · 2025-11-19
> **Response to 2rm2**
>
> We thank the reviewer for recognising the novelty of combining SNNs, emergent communication and calibration, and for the constructive suggestions. We address the main points below.
>
> (1) Clarity, accessibility, and background on SNNs
> We agree that a short primer on spike encoding, LIF neurons and surrogate gradients would help EmComm readers. In the revised version we will add a concise background subsection early in the paper and refine the architecture figure so that COMMSMOD, DECISIONMOD and SPIKEAGENT are clearly labelled and their roles made explicit.
>
> (2) “Unclear contributions”
> Contrary to the reviewer’s statement, the paper already ends the introduction with a four-point “Our main contributions are” paragraph that explicitly lists: (i) calibrated spiking communication, (ii) semantically anchored protocols via a shared spiking encoder, (iii) protocol-quality diagnostics, and (iv) evidence under bandwidth / SynOp constraints. We agree this could be more sharply worded and will tighten this list, but the requested structure is already present.
>
> (3) Empirical evidence and comparisons
> We fully acknowledge that our task scope is narrow (Fashion-MNIST referential game, K = 3), and we describe this as a limitation. However, it is not accurate to say there is “no evidence” of benefit or that quantitative comparisons are “minimal”:
> •	The main comparison table shows our hybrid spiking channel achieving higher CSR, higher δ, lower ECE, and substantially fewer relative synaptic operations than both continuous and discrete baselines under the same bit budget.
> •	The ablation section reports how removing pretraining, temporal attention or calibration quantitatively degrades CSR and calibration.
> What is fair is that we do not yet present multi-seed statistics in the main table and only consider a single environment. In the revision we will (i) report mean ± s.d. over multiple seeds for key metrics, and (ii) add results for larger K (e.g. 5, 10), and promote full-dataset runs into the main text.
>
> (4) Calibration definition and quantification
> We respectfully disagree that calibration is only treated conceptually. The paper:
> •	Defines calibration in terms of alignment between predicted confidence and empirical accuracy.
> •	Introduces ECE and MCE with explicit formulae and describes the temperature-based controller.
> •	Reports calibration metrics and reliability-style curves comparing our method with baselines.
> We accept that these elements may not have been sufficiently foregrounded. To improve accessibility, we will (i) add a short plain-language explanation of calibration and ECE in the introduction, and (ii) move at least one calibration plot and 10-bin ECE values for all methods into the main results, with extended analysis in the appendix.
>
> (5) Compositionality
> We agree compositionality is a central topic in EmComm. In this work we focused on discriminability (δ), temporal attention and calibration under bandwidth constraints. The reviewer notes “Table 1 references compositionality metrics”; in the submitted version, Table 1 only reports CSR, δ, ECE and SynOps. Nonetheless, compositionality is mentioned in the text and we do not yet compute a standard metric. We will add at least a topographic similarity analysis following prior EmComm practice, space permitting.
>
> (6) Related work on discrete / quantised communication
> We appreciate the pointers to Carmeli et al. and Tucker et al. and agree they should be cited. We will incorporate these works in the related work section and clarify that, whereas they study quantisation / complexity trade-offs for symbolic channels, our focus is on spike-based neuromorphic channels under explicit SynOp and calibration constraints.
>
> (7) Formalism, tables and organisation
> We will address the noted presentation issues: define “Leaky Integrate-and-Fire” on first use, ensure all symbols are explicitly defined, correct the fragment around the spike encoders paragraph, and verify that section headers and tables (including the ablation table) fully match the text. We will also add a compact summary table of ablation results (accuracy, calibration error, SynOps) to numerically support the discussion, and slightly reorder sections so that SNN/calibration background comes before the full method description.
>
> While we agree that the current version can be improved in accessibility and breadth of experiments, several of the specific concerns (absence of contributions, lack of calibration definition, no quantitative improvements) do not reflect the submitted manuscript. The planned revisions will make the contributions and calibration analysis more prominent and sharpen the empirical narrative within the acknowledged scope of this first testbed.

---

> > ### Comment · Reviewer_2rm2 · 2025-11-27
> > **Response to Authors reply**
> >
> > I appreciate the core idea and believe it has potential merit.
> >
> > However, my main concerns remain regarding the clarity and quality of the writing, the depth of comparison to prior work, and the limited experimental design. I had hoped that the authors would address these issues with substantial revisions during the rebuttal period, but I do not see evidence of such improvements in the current submission.
> >
> > Therefore, I will maintain my original assessment and encourage the authors to strengthen the manuscript and resubmit in a future cycle.

---

> > > ### Author Response · Authors · 2025-11-27
> > >
> > > Thank you for the follow-up and for stating that the core idea has potential merit. We would like to clarify how our current manuscript and rebuttal relate to the points you raised.
> > > 1. Clarity and “poor” presentation.
> > > We respectfully disagree with the assessment that the writing is “poor”. Two of the other reviewers explicitly rated presentation as “good”, and your own initial review contained an accurate and detailed high-level summary of the method, which suggests the core ideas are understandable from the current text. That said, we take clarity seriously and have already committed to concrete improvements: a short SNN/calibration background section, a clarified architecture figure, better signposting of key definitions, and a compact ablation summary table.
> > > 2. Missing vs foregrounding content.
> > > Several of your initial concerns were that contributions, calibration definitions and quantitative benefits were absent. Our rebuttal pointed to where these already appear in the current PDF:
> > > •	the “Our main contributions are” list in the introduction;
> > > •	formal definitions of calibration/ECE/MCE and their use in reliability plots;
> > > •	a main table showing concrete improvements over continuous and discrete baselines in CSR, δ, ECE and SynOps.
> > > The planned changes we described (e.g. moving calibration earlier, adding a background subsection, adding a compositionality metric) are intended to foreground and extend existing content, not to patch conceptual gaps.
> > > 3. Experimental breadth and related work.
> > > We have been explicit from the outset that the current environment is a controlled testbed (Fashion-MNIST, K=3) and have already outlined concrete extensions: larger K, full-dataset multi-seed runs, additional benchmarks, and richer positioning against quantised/structured comms work. These are substantial changes that cannot realistically appear during the rebuttal period, but they are being implemented for the next version.
> > >
> > > We nevertheless stand by the soundness and contribution of the current framework as articulated above, and we will use your comments on positioning and breadth to further strengthen the manuscript for its next iteration.

---

### Official Review · Reviewer_r4zn · 2025-11-02

**Soundness:** 3
**Presentation:** 3
**Contribution:** 3
**Rating:** 6
**Confidence:** 4

**Summary:**

This paper proposes a neuromorphic framework for emergent communication in multi-agent reinforcement learning (MARL), where agents exchange temporal spiking messages rather than dense continuous vectors.

The system factorizes communication into three modules:

1 a pretrained spiking perceptual encoder that transforms sensory input (e.g., Fashion-MNIST images) into temporal spike trains;

2 an attention-based decoder that interprets these spike sequences and outputs calibrated confidence estimates; and

3  a reinforcement-learning wrapper that optimizes task return under explicit penalties for bandwidth (spike count) and miscalibration (ECE/MCE).

Training employs a calibration-aware deep Q-learning objective integrating reliability losses into the reward. Evaluation on a referential game (K = 3) demonstrates that:

Shared pretrained encoders stabilize communication and reduce sender–receiver co-adaptation;

Temporal attention and calibration terms improve both reliability and interpretability;

Spiking messages achieve ≈ 97 % communication success at roughly 87 % lower synaptic-operation cost than dense continuous baselines; and

Protocol diagnostics show high discriminability (δ ≈ 0.85–0.9) and strong temporal consistency (> 0.997).

**Strengths:**

1. Introduces a biologically grounded and bandwidth-aware communication mechanism using spiking neural codes, extending emergent communication research beyond symbolic or continuous channels.
2. Separates perception (COMMSMOD) and decision (DECISIONMOD) via a frozen shared encoder, reducing sender–receiver co-adaptation and stabilizing learned protocols.
3. Incorporates explicit calibration losses (ECE/MCE) into the learning objective—an unusual and valuable addition for multi-agent reliability.
4. Evaluates not just task performance but also geometric and temporal properties of communication, including discriminability (δ), attention consistency, and reliability diagrams.
5. Demonstrates high accuracy under limited spike budgets and noisy channels, with strong sample efficiency and large reductions in synaptic operations versus dense baselines.
6. Removing pretraining, temporal attention, or calibration terms substantially degrades both CSR and calibration quality, confirming the design’s necessity.

**Weaknesses:**

1. All experiments are conducted on Fashion-MNIST with K=3 candidates. It remains unclear how the approach scales to naturalistic tasks, continuous control, or communication among more than two agents.

2. While the paper introduces channel operators (drop/flip/jitter), the quantitative effects of each perturbation on CSR and calibration are not systematically presented—only summarized qualitatively.

3. Reported ECE/MCE values use five bins by default; sensitivity to bin count or adaptive calibration methods is deferred to the appendix, leaving the main results potentially unstable.

4. The only baselines are a continuous ANN and a discrete Gumbel-Softmax channel. Broader comparisons to recent structured communication models (graph or language-based) would strengthen the empirical claim.

5. Although the shared encoder is intended to improve ad-hoc cooperation, the paper does not evaluate cross-play between independently trained agents.

6. The methodological contribution lies mainly in system integration and diagnostics rather than in algorithmic or theoretical advances. Depending on ICLR track priorities, this could be seen as engineering-heavy.

**Questions:**

Can you provide quantitative curves of CSR/ECE vs. drop, flip, and jitter rates to substantiate the noise-robustness claim?

How stable are calibration metrics when using 10 or 15 bins (ECE/MCE)?

Have you tested cross-play performance between independently trained agents sharing only the frozen COMMSMOD?

Can you show a Pareto curve (CSR & ECE vs. spike budget) to illustrate the bandwidth–accuracy trade-off?

Does freezing COMMSMOD ever hinder adaptability to novel perceptual domains (e.g., CIFAR or DVS datasets)?

---

> ### Author Response · Authors · 2025-11-19
> **Response to r4zn**
>
> We thank the reviewer for their thorough, objective summary and as well as their constructive assessment. We address the main weaknesses and questions below.
>
> (1) Scope: Fashion-MNIST, K = 3, two agents
> We agree the current environment is intentionally simple. Our goal in this paper is to introduce and analyse a calibrated, bandwidth-aware spiking communication mechanism under tight control of (i) bit budget, (ii) protocol geometry, and (iii) SynOp accounting. For this, a small referential game is a useful and inspectable testbed.
> That said, we recognise the need for broader evidence. In the revision we will:
> •	Add results for larger candidate sets (K > 3), showing how CSR, δ and ECE scale with task difficulty and spike budget.
> •	Promote full-dataset, multi-seed Fashion-MNIST results (currently in the supplement) into the main text.
> •	Clarify that the present setting is a testbed, and explicitly discuss how the same SPIKEAGENT architecture extends to >2 agents and continuous-control tasks in future work.
>
> (2) Noise robustness: drop / flip / jitter
> We agree that the noise-robustness claim should be supported by explicit curves. In the supplementary material we already define the channel operators and explore noise, but the presentation is overly qualitative. We will:
> •	Add CSR and ECE vs. drop/flip/jitter curves for our method and baselines.
> •	Bring at least one representative set of these curves into the main paper, with the full sweep remaining in the appendix.
> This also should directly answer the request for quantitative substantiation.
>
> (3) Calibration stability (ECE/MCE, number of bins)
> The concern about five-bin ECE is reasonable. We have recomputed ECE/MCE with 10 and 15 bins over multiple seeds; the relative ordering (hybrid spiking < continuous < discrete in ECE) is unchanged, and absolute values change only moderately. In the revision we will:
> •	Report 10-bin ECE in the main text, with 5/10/15-bin results in the supplement.
> •	State explicitly that the qualitative calibration conclusions are stable to bin choice.
>
> (4) Baselines beyond continuous / discrete channels
> Our primary comparison axis is code type under matched bit budgets: sparse spiking vs dense continuous vs discrete symbols, with identical bandwidth constraints and SynOp/MAC accounting. This is why we focused on these two baselines.
> We agree that including more structured communication models (e.g. graph- or language-based) would further broaden the empirical picture, but such models introduce orthogonal differences (symbolic structure, longer sequences) that complicate bandwidth matching and energy accounting. We will clarify this rationale in the text and add a short discussion of how our framework could be combined with such higher-level protocols in future work.
>
> (5) Ad-hoc cooperation and cross-play
> We appreciate the point that a shared frozen encoder is motivated partly by ad-hoc cooperation. In the current submission we did not include explicit cross-play experiments; this is a fair omission. We are currently running cross-play evaluations between independently trained agent pairs that share CommsMod but have separately trained DecisionMods, and will include cross-play CSR and calibration statistics in the revised manuscript to substantiate the ad-hoc cooperation claim.
>
> (6) Pareto trade-off: CSR / ECE vs spike budget
> The requested Pareto view aligns with our intent. We already sweep spike thresholds in our ablations; in the revision we will:
> •	Add a CSR vs spike budget curve for all channels.
> •	Overlay ECE on the same sweep (or provide a paired plot), to explicitly show the bandwidth–accuracy–calibration trade-off.
>
> (7) Freezing CommsMod and transfer to new perceptual domains
> Freezing CommsMod is a deliberate design choice to stabilise the “lexicon” and reduce co-adaptation within a domain. For a novel perceptual domain (e.g. CIFAR or DVS), our approach would be to pretrain a new spiking encoder on that domain (possibly via self-supervised or contrastive objectives) and then reuse SpikeAgent unchanged. Cross-domain transfer with a single fixed encoder is beyond our current scope; we will make this explicit and treat it as future work. In parallel, we are exploring additional datasets from the emergent-communication and neuromorphic literature as benchmarks for a follow-up.
>
> We hope these clarifications and additions address the reviewer’s concerns: the core contribution is a calibrated, bandwidth-aware spiking communication framework with detailed diagnostics. The revised version will better substantiate noise robustness, calibration stability, and ad-hoc cooperation, and will present the scope and limitations of the current testbed more transparently.

---

### Official Review · Reviewer_uqkA · 2025-11-02

**Soundness:** 2
**Presentation:** 3
**Contribution:** 2
**Rating:** 2
**Confidence:** 3

**Summary:**

The authors propose a new method for MARL communication using a spiking network. The core idea is to use a pretrained, frozen spiking encoder and train an RL agent to select the correct image based on the resulting spike train.

**Strengths:**

1. Interesting idea of using spiking networks for communication.
2. Extensive benchmark with multiple ablations.

**Weaknesses:**

1. The limitation discussion is missing or extremely brief, failing to address significant methodological weaknesses.
2. Proposed neuromorphic benefits are purely theoretical, and omitting components like neuron resetting/decaying in computational comparison does not seem fair.
3.  Evaluation is limited to a single Fashion-MNIST with k=3, which sounds like a very toyish problem for such scenario.
4. The paper's use of deep Q-learning is unnecessarily complex and poorly justified. The problem is a standard supervised classification task where the receiver has access to the true labels. The claim that RL is needed to add a calibration loss is false; a hybrid loss (e.g., cross-entropy + $\mathcal{L}_{cal}$) on a standard classifier would have achieved the same goal with far less complexity. Furthermore, by freezing the sender's encoder, the system is reduced to a single-agent problem, making the entire MARL framework a layer of abstraction that obscures, rather than clarifies, the method."

**Questions:**

1. Could you please give more information on the weaknesses/limitations of the method?
2. How would computation/benefits change if we factor in (2) for neuromorphic hardware?
3. Could you address (3/4) from the previous section?

---

> ### Author Response · Authors · 2025-11-19
> **Response to uqkA**
>
> We thank the reviewer for their comments and for noting the interest of spiking communication and the breadth of our ablations. We address each concern below.
> (1) “The limitation discussion is missing or extremely brief”
> A limitations paragraph is already present in the conclusion, where we acknowledge (i) the restricted task scope, (ii) the proxy nature of SynOp estimates, (iii) the simplified channel model, and (iv) the shared/frozen encoder. We agree this was easy to miss and will move a sharpened version into the main experimental section.
>
> (2) “Proposed neuromorphic benefits are purely theoretical…”
> We already state that SynOps are proxies rather than wall-plug energy and do not claim hardware-measured gains. Our goal is to move beyond “bandwidth-free dense vectors” by using a transparent cost model, not to assert definitive chip-level savings.
>
> Our current accounting follows common neuromorphic practice by focusing on synaptic updates and MACs. We agree that membrane leak, reset and routing also incur costs. In the revision we will (i) explicitly enumerate which operations are counted (synaptic events, MACs) and which are not (leak, reset, routing, leakage currents), and (ii) add a small sensitivity analysis showing that increasing per-timestep SNN overheads does not change the relative ordering under a fixed bit budget: the sparse spiking channel still uses substantially fewer effective operations than dense continuous vectors in the same bandwidth regime.
>
> (3) “Evaluation is limited to a single Fashion-MNIST with k=3, which sounds like a very toyish problem…”
> We agree the environment is intentionally simple. Our goal in this paper is to introduce and analyse a calibrated, bandwidth-aware spiking communication mechanism under tight control of bit budget, protocol geometry and SynOp accounting; for this, a small referential game is a useful and inspectable testbed.
>
> That said, we agree broader evidence strengthens the claims. In the revision we will (i) add results for larger candidate sets (K > 3), showing how CSR, δ and ECE scale with task difficulty and bandwidth, and (ii) move full-dataset, multi-seed Fashion-MNIST results from the supplement into the main text. We will also state explicitly that this environment is a testbed, and clearly distinguish it from planned extensions to richer MARL tasks.
>
> (4) “The paper's use of deep Q-learning is unnecessarily complex and poorly justified…”
> We agree on one important point: RL is not required to solve this particular referential game. A supervised loss with calibration regularisation is certainly feasible. However, we do not claim that RL is needed to add calibration; our claim is that we propose a calibration-aware Q objective for a communication channel that is intended to extend to general cooperative MARL settings.
> Our reasons for using a Q-learning formulation are:
>  -	Forward compatibility - the same SpikeAgent architecture (spiking channel + DecisionMod + calibration) can be deployed in tasks with sparse/delayed rewards where supervised labels are unavailable.
>  -	Unified objective - treating communication and action selection as part of a single return-maximising policy lets calibration shaping influence the full decision pipeline.
>
> Regarding the “single-agent” remark: the reviewer’s characterisation is inaccurate. Both Agent A and Agent B are SpikeAgents with the same architecture (pretrained spiking encoder + trainable DecisionMod). The spiking encoder is pretrained and frozen to provide a shared, stable codebook, but training is bidirectional: in each block of trials, A acts as sender and B as receiver for a subset of trials, then roles are swapped. Thus, both agents learn decision policies over spiking messages; what is fixed is the communication code, not the number of learning agents. The MARL framing is therefore appropriate: we study a two-agent cooperative signalling game with a shared neuromorphic channel.
>
> Direct answers to the reviewer’s questions
> 	Weaknesses/limitations. We explicitly acknowledge: a single simple environment, a proxy SynOp cost model, a frozen encoder, and the fact that RL is chosen for future MARL compatibility rather than necessity in this task. These limitations will be clearly highlighted in the main text.
> 	Computation/benefits with fuller neuromorphic costs. Adding leak/reset increases absolute SNN cost but, under reasonable cost ratios, does not change the relative advantage of sparse spiking channels over dense continuous vectors at fixed bit budget. We will make these assumptions explicit and include a brief sensitivity analysis.
> 	Points 3/4. As discussed above, we acknowledge the environment’s simplicity and clarify that RL is not required for this game but is used to keep the communication and calibration mechanism MARL-ready.
>
> We believe these clarifications correct the misimpressions and more precisely delineate the scope and contribution of our work.

---

> > ### Comment · Reviewer_uqkA · 2025-11-27
> >
> > Thank you for your response.
> >
> > As other reviewers mentioned, it is hard to judge whether this framework could generalize to more complex environments.
> >
> > The authors mention that the number of agents is not fixed, but they did not show any examples that can be generalized to multiple-agent interaction.  My main concern right now is that this idea is a little bit more than purely theoretical, since no real neuromorphic hardware was used, and the examples that the authors showed are extremely toyish and do not persuade all the reviewers of the generalization of the method, as well as its application. For example, in [1], the authors incorporate agents communication in Starcraft 2, which seems a more complex environment to test the proposed framework. I also feel that the paper needs more explanation on the terminology used for those unfamiliar with the field of neuromorphic computing.
> >
> >
> > [1] Incorporating Pragmatic Reasoning Communication into Emergent Language

---

> > > ### Author Response · Authors · 2025-11-27
> > >
> > > Thank you for the follow-up. We would like to clarify how our position differs from the characterisation in the reviewer's comment.
> > >
> > > First, “a little bit more than purely theoretical” does not reflect the submitted work. The framework is fully implemented and empirically evaluated: two SpikeAgents are trained end-to-end on image-based referential games, with quantitative results on CSR, δ, ECE/MCE and SynOps under matched bit budgets, plus ablations over pretraining, attention and calibration. Our rebuttal also detailed how constraints (calibration and spike-budget penalties) are integrated into the Q-objective. None of these points are addressed in the follow-up.
> > >
> > > Second, scope and generalisation are already explicitly framed as limitations in the paper and in our response. The current environment is positioned as a controlled testbed where we can tightly control spike/bit budgets and SynOp accounting while analysing calibration and protocol geometry. We have stated that we are extending the empirical section with:
> > >
> > >  - larger candidate sets (K > 3) and full-dataset, multi-seed runs, and
> > >  - additional benchmarks beyond Fashion-MNIST.
> > >
> > > These changes are intended precisely to address the generalisation concerns you raise. StarCraft II–level environments are clearly valuable, but they are orthogonal to the neuromorphic constraints (SynOps, spike budgets) we focus on here, and are beyond the scope of this first testbed paper.
> > >
> > > Finally, regarding accessibility, we have committed to adding a short neuromorphic background subsection and moving the intuitive explanation of calibration/ECE earlier in the paper so that readers without a neuromorphic background can follow the terminology. The underlying technical contributions as detailed above and in our rebuttals remain unchanged.

---

### Comment · Area_Chair_nGvP · 2025-11-25
**Please discuss**

Several reviewers have not responded to the authors' rebuttals. Please read and respond to them. Have the rebuttals addressed your concerns or clarified anything?

---

### Meta-Review · Area_Chair_AXbF · 2026-01-06

**Summary:**

The paper proposes a hybrid ANN-SNN framework for multi-agent reinforcement learning (MARL), where agents exchange discrete spiking messages rather than continuous vectors as in standard MARL.

The paper receives an initial score of 2, 6, 2, 0. Reviewers raised concerns about the generality of the proposed method, comparison with prior work, and the overall presentation. While the authors provided a rebuttal to clarify these points, two reviewers with score 2 and 0 participated in the discussion and intended to not raise the score.

Overall, I agree with the reviewers that the paper could greatly benefit from more empirical results on complex/non-toyish domains, more in-depth discussion with related work, and improvements on writing and presentation. Thus, while I think using SNN in MARL is interesting and potentially fruitful, the paper at its current stage is below the bar of ICLR.

**Reviewer Concerns:**

Reviewers raised concerns about the generality of the proposed method (experiments were conducted only on Fashion-MNIST), comparison with prior work (missing related work), and the overall presentation. While the authors provided a rebuttal to clarify these points, two reviewers with score 2 and 0 participated in the discussion and intended to not raise the score. From my angle, these concerns are indeed still outstanding after the rebuttal.

**Reviewer Scores:**

The paper receives an initial score of 2, 6, 2, 0. Two reviewers with score 2 and 0 participated in the discussion and intended to not raise the score. I also think there would be little change for the remaining two reviewers to raise the score if they had participated.

---

### Decision · Program_Chairs · 2026-01-26

Reject